# Mechanisms of Resistance and Strategies to Combat Resistance in PD-(L)1 Blockade

**John Moise [1], Jeevan Murthy [1], Dolma Dabir [2,3], Stephen Yu [2], Farah Kisto [1], Emily Herron [1] and Sonikpreet Aulakh [2,3,\*]**

[1]   West Virginia University School of Medicine, Morgantown, WV 26506, USA
[2]   Department of Medicine, West Virginia University School of Medicine, Morgantown, WV 26506, USA
[3]   Department of Neuroscience, West Virginia University School of Medicine, Morgantown, WV 26506, USA
\*   Correspondence: sonikpreet.aulakh@hsc.wvu.edu; Tel.: +1-304-598-6984

**Abstract:** Prolonged survival and durable responses in several late-stage cancers such as melanoma and lung cancer have been made possible with the use of immune checkpoint inhibitors targeting the programmed cell-death protein 1 (PD-1) or its ligand PD-L1. While it is prudent to focus on the unprecedented and durable clinical responses, there are subsets of cancer patients that do not respond to immunotherapies or respond early and then relapse later. Many pathways of resistance have been characterized, and more continue to be uncovered. To overcome the development of resistance, an in-depth investigation is necessary to identify alternative immune receptors and signals with the overarching goal of expanding treatment options for those with demonstrated resistance to PD1 checkpoint immunotherapy. In this mini-review, we will discuss the mechanisms by which tumors exhibit resistance to anti-PD-1/PD-L1 immunotherapy and explore strategies to overcome such resistances.

**Keywords:** cancer; immunotherapy; PD1; PDL1; resistance

## 1. Introduction

Immune checkpoint inhibition is a promising mode of immunotherapy that has shown incredible clinical efficacy in the treatment of several types of solid and liquid tumors. Immune checkpoint inhibitors (ICIs) are monoclonal antibodies that physically interfere with the transduction of inhibitory signals to activated T-cells, thereby potentiating a robust anti-tumor immune response [1]. Of the nine ICIs that have received FDA approval for clinical use, seven target the PD-1/PD-L1 immune checkpoint–an approach that has shown to be strikingly efficacious while exhibiting acceptable patient tolerability profiles [2–4].

Programmed cell death protein 1 (PD-1) is a transmembrane receptor protein first identified in 1992 during a screen for apoptosis-associated molecules by now-Nobel-laureate Tasuku Honjo and others at Kyoto University. Its function remained unknown until 1999, when the same group of researchers used murine models to show that PD-1 is an important negative regulator of immune responses and that its deficiency leads to the development of organ-specific autoimmunity [5,6]. PD-1's ligand, PD-L1, was identified later that same year alongside experimental evidence confirming that the PD-1/PD-L1 interaction is indeed what mediates self-tolerance through its transduction of inhibitory signals to activated T cells [7,8]. PD-L1 expression plays a critical role in preventing autoimmunity; however, some cancers co-opt this immune checkpoint and overexpress PD-L1 to induce immune tolerance locally within the tumor microenvironment (TME), thereby allowing the tumor cells to evade elimination by the immune system [1,9–12].

Inhibition of the PD-1/PD-L1 immune checkpoint with monoclonal antibody inhibitors has revolutionized the therapeutic landscape of several cancers, including melanoma, non-small cell lung cancer, renal cell carcinoma, and urothelial cancers [13]. However, despite the robust response rates, prolonged overall survival, and prolonged progression-free

survival, resistance to anti-PD-1/PD-L1 therapy is a significant challenge, as only a minority of patients (20–30%) are estimated to experience a positive response to PD-1/PD-L1 blockade therapy, while the rest experience cancer progression due to either primary or acquired resistance [14,15]. Further, while it has been observed that the clinical efficacy of PD-1/PD-L1 inhibitors correlates positively with the tumor mutational burden (TMB) and PD-L1 expression, the lack of uniformity in the expression-dependent effect suggests that there are additional mechanisms at play. The purpose of this mini-review is to elucidate the mechanisms underlying resistance to PD-1/PD-L1 checkpoint inhibition therapy and investigate emerging strategies for overcoming such resistance.

## 2. PD-1/PD-L1 Expression and Function

PD-1 is a transmembrane receptor protein that is expressed on the surface of activated T-cells and functions in attenuating the immune response by inhibiting the activity of effector T-cells [16]. Upon binding its ligand, PD-L1, the tyrosine residues on the cytoplasmic portion of the PD-1 protein become phosphorylated and bind Src homology region 2 domain-containing phosphatase-1 and 2 (SHP-1 and SHP-2), which proceed to dephosphorylate nearby signaling molecules, causing a profound downregulation in pathways responsible for T-cell metabolism, cytokine synthesis, proliferation, and survival, in addition to interfering with transcription factors, such as GATA-3, T-bet, and Eomes, that are implicated in T-cell effector functions [17,18].

Its ligand, PD-L1, is ubiquitously expressed on cells throughout the body, although its expression is differential. In certain tissues, such as the retinal pigment epithelium, pancreatic islets, and neurons, its expression is constitutive and serves to persistently protect these tissues from inadvertent damage caused by activated T cells [19,20]. In other tissues throughout the body, PD-L1 expression has been found to be inducible in the presence of a number of pro-inflammatory cytokines, such as interferon gamma (IFN-γ), which are released by tissue-infiltrating effector T-cells [16,20–22]. The pro-inflammatory cytokines cause a localized induction of PD-L1 expression that, through binding PD-1, serves to protect host tissue that is experiencing active infection and inflammation by markedly affecting the effector activity of both CD8+ and CD4+ effector T-cells. Activation of the PD-1 receptor on effector CD8+ T-cells induces T-cell exhaustion, resulting in a significant inhibition of effector activity, a reduction in proliferation, and a decrease in cytokine production [8,23]. At the same time, PD-1 receptors on CD4+ T-cells play a critical role in the differentiation and establishment of Treg cell populations, which attenuate the immune state and prevent autoimmunity [24].

## 3. PD-L1 Overexpression in Tumors and PD-1/PD-L1 Immune Checkpoint Therapy

PD-L1 is overexpressed in several malignancies, allowing the tumor cells to evade elimination by the immune system through inducing localized immune tolerance within the tumor microenvironment [1,9–12]. Higher expression levels of PD-L1 correlate strongly with a poorer prognosis by suppressing the activation of T cells, leading to tumor progression [25–31]. Recognizing this role of the PD-1/PD-L1 axis, several anti-PD-1/PD-L1 monoclonal antibody therapies have been developed to bind either the receptor or ligand to prevent the transduction of the inhibitory signals to the effector T-cells, thereby allowing the immune system to mount a robust anti-tumor response unimpeded by the tumor's PD-L1 expression levels [5,16,32–35]. The first of these ICIs were nivolumab (anti-PD-1) and pembrolizumab (anti-PD-1), both of which received FDA approval in 2014. Several additional agents targeting the PD-1/PD-L1 axis have since been approved, including atezolizumab (anti-PD-L1) in 2016, avelumab (anti-PD-L1) in 2017, durvalumab (anti-PD-L1) in 2017, cemiplimab (anti-PD-1) in 2018, and most recently dostarlimab (anti-PD-1) in 2021 [4].

PD-1/PD-L1 checkpoint inhibitor therapy has shown remarkable clinical efficacy in treating non-hematologic and hematologic malignancies. These treatments have become the standard of care for a select subset of neoplasms, most notably non-small-cell lung

cancer (NSCLC) and metastatic melanoma [36]. Other clinical indications of PD-1/PD-L1 checkpoint inhibitors have continued to diversify in our ongoing pre-clinical and clinical trials [37]. Recently, a phase 2 clinical study by Cereck et al. has shown remarkable success with Dostarlimab, a PD-1/PD-L1 checkpoint inhibitor, in locally invasive, mismatch repair-deficient adenocarcinomas of the colon. After more than six months of treatment, all 12 patients enrolled in the study achieved a clinically complete response, as confirmed by 18F-fluorodeoxyglucose positron emission tomography (FDG PET), a digital rectal exam (DRE), T2-weighted magnetic resonance imaging (MRI), endoscopy, and/or biopsy at baseline, and at 6 weeks, 3 months, 6 months, and every 4 months following. No further surgery or radiation or chemotherapy was required in these patients, indicating that cancers with these molecular features are highly sensitive to PD-1/PD-L1 immunotherapy [38]. In light of the successes of PD-1/PD-L1 checkpoint inhibition, it is important to note the emerging recognition of the importance of PD-L2, which has an equivalent function to PD-L1 in immune regulation. Current research, such as that of Yearly et al., suggests that anti-PD-1 monoclonal antibodies such as Nivolumab derive benefit through the simultaneous blocking of PD-1/PD-L1 and PD-1/PD-L2 interactions, in addition to suggesting that the use of combination therapy targeting both PD-1 ligands in lieu of monotherapy targeting PD-L1 alone may provide greater clinical benefit, particularly in tumors with high PD-L2 expression [39,40].

## 4. Mechanisms of Resistance to PD-1/PD-L1 Blockade and Corresponding Therapeutic Strategies

Despite the clinical successes seen with PD-1/PD-L1 immune checkpoint therapy, patient response to therapy is heterogeneous and unpredictable, in terms of both the length and magnitude of therapeutic efficacy. An estimated 70–80% of patients experience cancer progression due to primary or secondary resistance mechanisms that prevent the immune system from mounting a robust and persistent anti-tumor response [14,15,41]. These resistance pathways have been an active area of research, and efforts are in place to discover innovative therapeutic strategies to overcome PD-1/PD-L1 resistance, of which the most notable are outlined in this section.

### 4.1. Immunosuppressive Cytokine Pathways

Regulatory T-cells (Treg) are a subpopulation of suppressor CD4+T-cells that are involved in the suppression of immune-mediated tissue destruction towards self-antigens, helping prevent the development of autoimmunity [42]. However, Treg cells support tumor progression by secreting inhibitory cytokines such as Transforming Growth Factor-β (TGFβ), Interleukin-10 (IL-10), and IL-35 inhibiting the anti-tumor immune response of effector T cells, natural killer (NK) cells and dendritic cells (DCs) resulting in tumor progression [43]. Targeting these immunosuppressive cytokines overcomes ICI resistance by increasing effector T cells and NK cell infiltration and by decreasing the myeloid-derived suppressor cell (MDSC) population within the TME [42]. It is therefore hypothesized that the modulation of certain cytokines, such as chemokine ligand 12 (CXCL12), TGFβ, and IL-17A, using monoclonal antibodies could be an avenue to potentiate the efficacy of PD-1/PD-L1 blockers by surpassing primary or acquired resistance in these therapy-refractory cancers [44–47].

CXCL12 is a cytokine that is localized in tumor cells and binds to chemokine receptor 4 (CXCR4) to activate multiple signaling pathways, including mitogen-activated protein kinase (MAPK), wingless-related integration site (wnt), and epithelial growth factor/receptor (EGF/EGFR) for cell cycle progression and migration [48,49]. Through utilizing an antagonist to inhibit CXCL12 binding to CXCR4 in pancreatic cancer murine models, one study showed a marked increase in effector T-cell accumulation in the tumors and a reduction of tumor cell volume despite the continued presence of Treg cells. These results suggest that by neutralizing CXCL12 signaling, Treg-induced inhibitory pathways in tumors can be bypassed, allowing the recruitment of T-cells and thereby re-sensitizing immune cells and

tumor cells to PD-1/PD-L1 checkpoint inhibition, as illustrated in Figure 1 [48]. Similar results were seen in pre-clinical studies where CXCL12 antagonism in metastatic breast cancer led to increased T-cell infiltration and improved immunotherapeutic efficacy in murine models [50]. With the intent of clinical translation, clinical trials have combined CXCL12 antagonist and anti-PD-1/PD-L1 therapies and have shown these treatments to be safe and efficacious in metastatic pancreatic cancer and colorectal cancer [51,52]. In these studies, combination therapy consisting of CXCL12 antagonists and ICIs showed synergism and enriched TME with effector T-cells to generate a robust anti-tumor response.

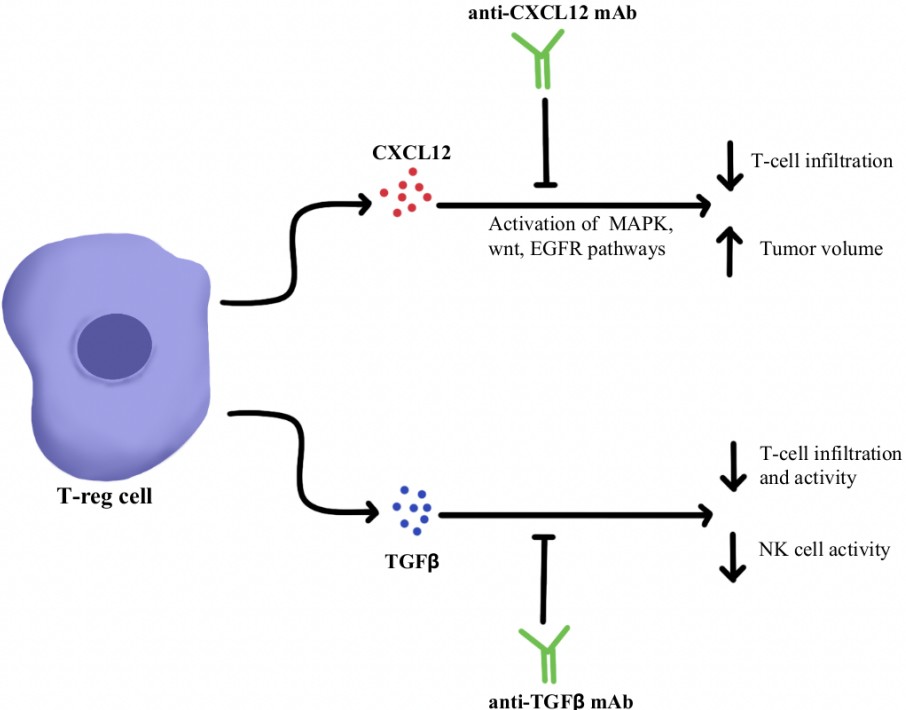

**Figure 1.** Diagram of CXCL12 and TGFβ cytokine pathway modulation [48–59]. Illustrated by Kaitlyn Mi of The Dartmouth Institute.

TGFβ is a cytokine that drives numerous cell processes, ranging from proliferation, motility, apoptosis, differentiation, and immune regulation [53]. TGFβ signaling in the immune TME, as illustrated in Figure 1, represses the anti-tumor activity of the immune cell population, including T cells and NK cells, resulting in immunosuppression that further limits the efficacy of ICIs and, as a result, induces therapeutic resistance to ICIs [54]. Studies using TGFβ inhibitors have been shown to increase T cell infiltration, Thelper1 (Th1)-type immune response, upregulation of Interferon-γ (IFN-γ), and promoted the lytic function of tumor antigen-specific CD8+T cells [55]. Further studies using a dual approach by simultaneously blocking the PD-1/PD-L1 checkpoints as well as the TGFβ pathway were successful in mounting a significant anti-tumor response by overcoming the ICI resistance as well as enhancing the recruitment of T cells [55–57]. The clinical data of TGFβ pathway inhibitors alone and in combination with ICIs have been studied in multiple advanced treatment-refractory cancers including pancreatic, cervix, colorectal carcinoma, biliary tract cancer, urothelial carcinoma, and NSCLC, and have shown to be safe, well-tolerated and provide better clinical outcomes (NCT03631706, NCT03732274) [58,59].

IL-17A is a proinflammatory cytokine that is produced by various cells, including Th17 cells, CD8+T-cells, γδT-cells, and NK cells within the TME [60]. IL-17A is expressed across different tumor types; however, the impact of IL-17A appears to be different based on the type of cancer. For instance, IL-17A serves as a tumor suppressor in melanoma and ovarian cancer, while in other cancers such as colorectal cancer and non-small cell lung cancer, IL-17A supports tumor growth [45,61–63]. Looking specifically at the impact of IL-17A and

its relationship to checkpoint inhibitors, IL-17A may potentiate the recruitment of MDSC and increase the immunosuppressive activity of Tregs within the TME that ultimately leads to resistance to anti-PD-1/PD-L1 therapy, as illustrated in Figure 2 [64–66]. Preclinical evidence suggests that IL-17A can promote migration and invasion of cancer cells by upregulating matrix metalloproteinase, and inhibition of a nuclear factor-kb (NF-kb) abolishes the tumor-promoting effects of IL-17A. These laboratory observations have not yet been translated into clinical applications. Due to the differential effects of IL-17A seen in different cancer types, the exact role of IL-17A in protective and aberrant processes has yet to be fully elucidated; however, this cytokine remains a possible modulation target to immunopotentiate the TME in ICI-refractory tumors [67].

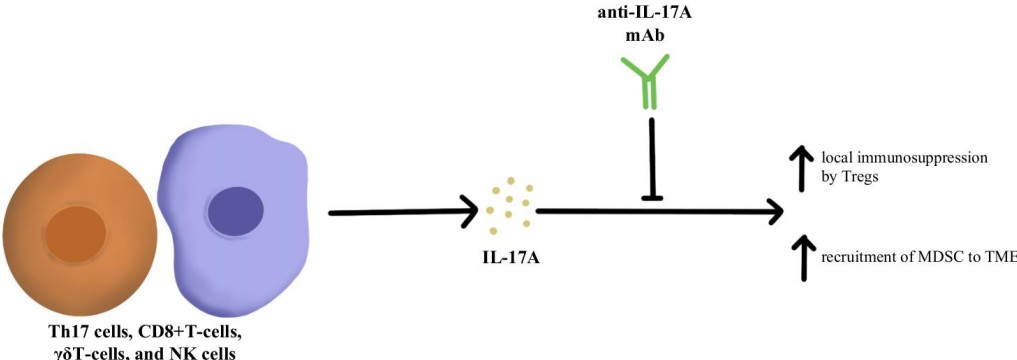

**Figure 2.** Diagram of IL-17A cytokine pathway modulation [45,60–67]. Illustrated by Kaitlyn Mi of The Dartmouth Institute.

### 4.2. T-Cell Exhaustion and Depletion

T-cell exhaustion within the TME is a significant mechanism of resistance to anti-PD-1/PD-L1 therapy. Exhausted T cells have been found to upregulate multiple inhibitory receptors such as cytotoxic T-lymphocyte-associated protein 4 (CTLA4), T-cell immunoglobulin, mucin-3 (TIM-3), lymphocyte activation gene 3 (LAG-3), and the T-cell tyrosine-based inhibitory motif (ITIM) domain [68–72]. Mutation of the ITIM domain affects PD-1 signaling and T-cell functional activity [73]. Overexpression of the T-cell immunoreceptor with immunoglobulin and ITIM domains (TIGIT) and increased infiltration of CD8+TIGIT+T-cells have been noted as correlating with worse prognosis in several cancers. Studies have shown that dual blockade of TIGIT/PD-1 blockers can overcome the T-cell dysfunction/exhaustion pathway of resistance [74]. CTLA4, discovered by 2018 Nobel Prize co-recipient James Allison, is an immune checkpoint protein expressed on activated T-cells that binds either CD80 and/or CD86 found on antigen-presenting cells (APCs) and regulates T-cell activity and effector functions [75]. CTLA4 and PD-1/PD-L1 pathways interact with each other in cancer in a complex manner, and studies have clearly shown the benefit of combining anti-CTLA4/anti-PD-1 treatments, both in preclinical and clinical settings, to reverse T-cell exhaustion [76]. LAG-3 interacts with major histocompatibility-II (MHC-II) to prohibit the binding of MHC molecules to T-cell receptor (TCR) and CD4+T cells, therefore, directly hindering TCR signaling in immune response [77]. A combination of anti-LAG-3 antibodies with nivolumab in a clinical study has shown it to be a safe and efficacious regimen in immunotherapy-resistant melanoma through its modulation of this pathway, as can be seen in Figure 3 [78].

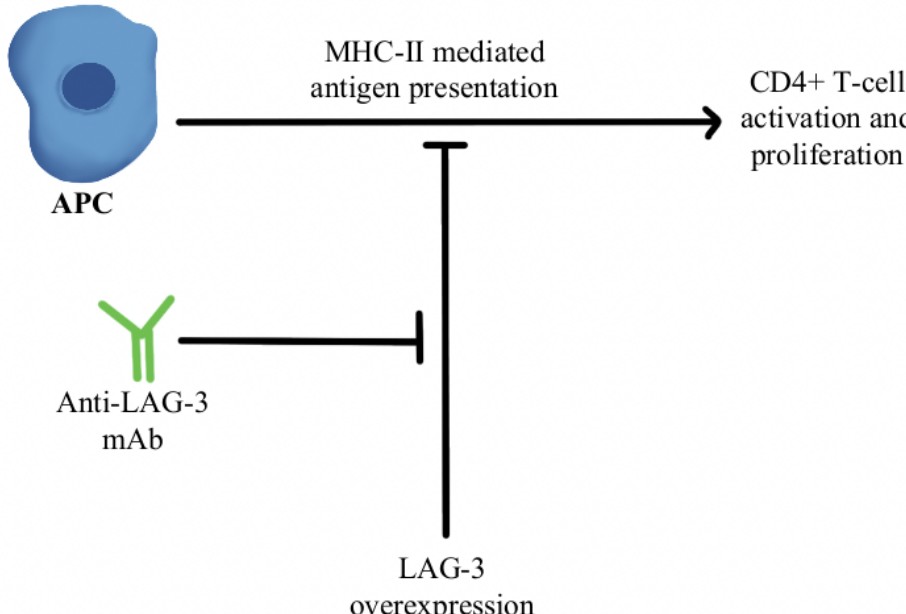

**Figure 3.** Diagram of LAG-3 pathway modulation [77,78]. Illustrated by Kaitlyn Mi of The Dartmouth Institute.

Another significant mechanism of resistance is the upregulation of TIM-3 on dysfunctional tumor-infiltrating lymphocytes (TILs), leading to an exhaustion phenotype [79]. It has been shown that over-expression of TIM-3 on Treg cells is associated with the development of resistance to anti-PD-1/PD-L1 therapy in melanoma [80]. The use of TIM-3 antibody in combination with chemotherapy indirectly enhanced CD8+T-cell responses as well as increased chemokine ligand 9 (CXCL9) expression by DCs [81]. TIM-3 mediates the exhaustion of T-cells and escape of PD-1 inhibition by activating phospho-inositol-3 kinase (PI3K)/protein kinase B(Akt) signaling [82]. In phosphatase and tensin homolog (PTEN) deleted tumors, PI3K/AKT pathway enhances the expression of PD-L1 and inactivates T-cells. Further, a selective PI3Kβ inhibitor in murine cancer models has revealed improvements in the efficacy of both anti-PD-1 and anti-CTLA-4 antibodies [83]. Additionally, a clinical study combining anti-TIM-3 antibody with anti-PD-1 immunotherapy in 219 patients was deemed safe and efficacious in solid malignancies [84]. There is also an ongoing clinical trial, slated to complete in December of 2022, that is investigating the modulation of TIM-3′s ligand, galectin-9, in potentiating the efficacy of anti-PD1 therapy in the treatment of solid tumors (NCT04666688). Figure 4 contains a diagram of this pathway and its modulation targets.

Another reported mechanism of resistance to PD-1/PD-L1 checkpoint inhibitors is increased levels of collagen in the tumor extracellular matrix that activates leukocyte-associated immunoglobulin-like receptor (LAIR-1), a membrane-bound receptor expressed by CD8+T-cells following CD18 interaction with collagen, causing a signaling cascade that induces T-cell exhaustion through SHP-1 signaling [85]. Preliminary in vivo studies have shown that LAIR-2 protein, a soluble receptor with greater affinity for collagen than LAIR-1, regulates the immune system by preventing LAIR-1 cross-linking to collagen via competitive binding [86]. Experiments are currently underway using LAIR-1 fusion proteins to probe for potential ligands for this inhibitory receptor for pre-clinical and clinical use, as diagrammed in Figure 5 [87].

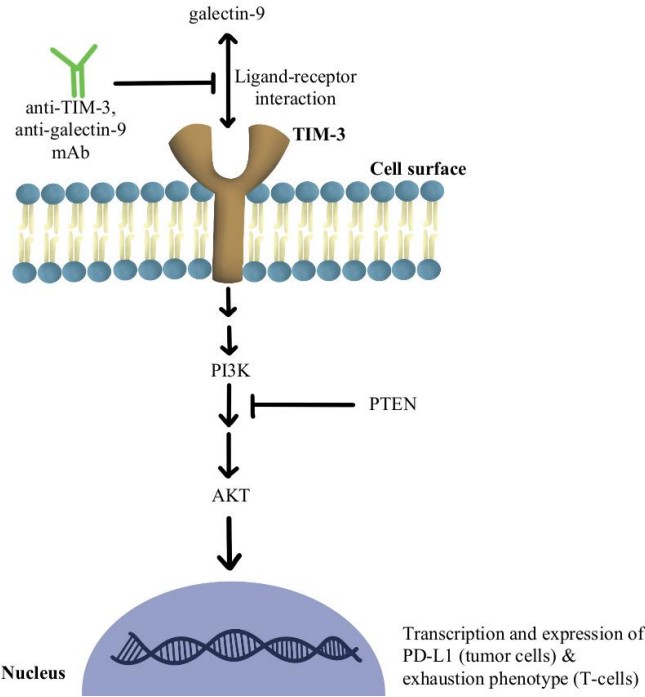

**Figure 4.** Diagram of TIM-3 pathway modulation (NCT04666688) [79–84]. Illustrated by Kaitlyn Mi of The Dartmouth Institute.

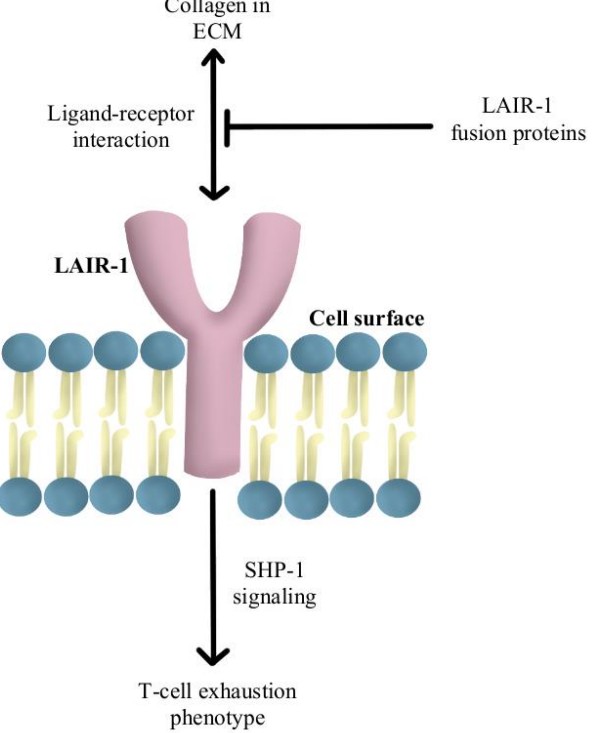

**Figure 5.** Diagram of LAIR-1 pathway modulation [85–87]. Illustrated by Kaitlyn Mi of The Dartmouth Institute.

T-cells recognize antigens presented by APCs. However, mutations in beta-2-microglobulin (B2M) can disrupt antigen presentation that ultimately leads to immune checkpoint blockade therapy resistance [88,89]. Whole exome sequencing of immunotherapy-resistant tumors has shown that truncating mutations in B2M and mutations in the genes encoding interferon-receptor-associated Janus kinase 1/2 (JAK1/2) result in a lack of response to

IFN-γ with loss of surface expression of major histocompatibility complex class I [90]. With this loss of MHC class I, tumors with the B2M mutation respond to anti-PD-1 therapies through the activation of MHC class I-independent mechanisms mediated by CD4+T-cells or NK cells [91]. Bempegaldesleukin, a CD122-preferential IL-2 pathway agonist, has been shown to activate CD4+T-cells and NK cells regardless of the PD-1/PD-L1 status of tumors [92]. Additionally, when Bempegaldesleukin was used in combination with immunotherapeutics, including anti-PD-1 and anti-CTLA4, it resulted in intratumoral depletion of Treg cells and increased proliferation of CD8+T-cells [93]. As the mutations in JAK1/2 led to resistance to PD-1 blockade therapy, a preclinical study has shown that administration of Toll-like receptor 9 (TLR9) agonists can overcome JAK1/2 knockout-induced resistance mediated by NK and CD8+T-cells [94]. Following these observations, CMP-001 (a TLR9 agonist) has been developed for study in a phase Ib clinical study in combination with Pembrolizumab in advanced melanoma (NCT02680184). Another phase II study evaluated the combination of IMO (another TLR9 agonist) in combination with ipilimumab (anti-CTLA4) in immunotherapy-refractory melanoma and has shown a revival of immune responses in 15 out of 24 enrolled patients [95]. These drug combinations and their respective effects on the MHC-I-independent and MHC-I-dependent pathways are illustrated in Figure 6.

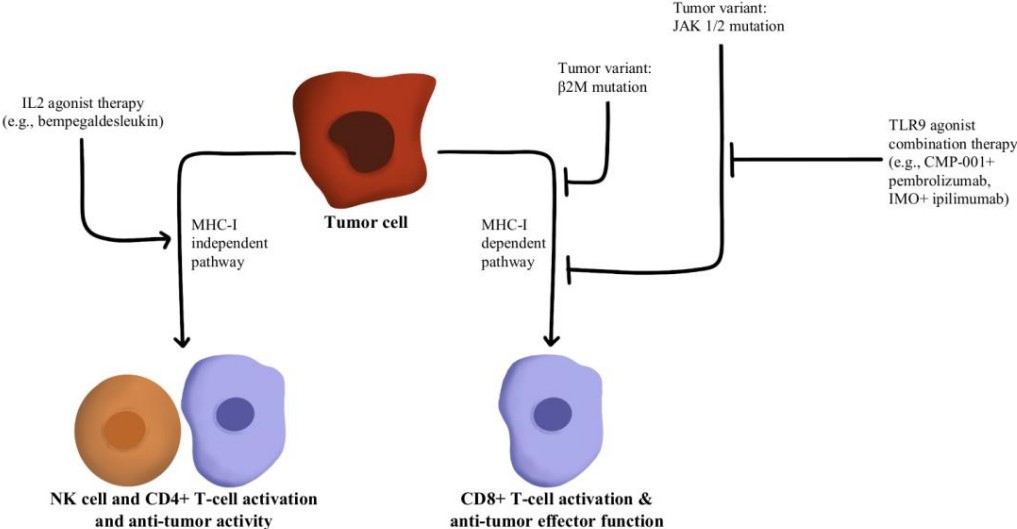

**Figure 6.** Diagram of MHC-I-dependent and MHC-I-independent pathway modulation [88–95]. Illustrated by Kaitlyn Mi of The Dartmouth Institute.

Depletion of T-cells within the TME is facilitated by chronic, persistent type II interferon signaling, enabling signal transducer and activator of transcription (STAT) tumor-related epigenetic changes, resulting in increased expression of interferon-stimulated genes and inhibitory receptors (TCIRs) on multiple T-cells, including Galectin-9, MHC-II ligands, and immune inhibitory checkpoints, including TIM-3 and LAG-3. Increased co-expression of multiple TCIRs aggravates T-cell depletion, while blocking interferons can reverse resistance caused by T-cell depletion [96,97]. Konen and others have found that neurotrophic tyrosine receptor kinase (NTRK) is upregulated by anti-PD-1 therapy. NTRK also activates the JAK-STAT signaling pathway and by upregulating the expression of multiple inhibitory receptors on T-cell surfaces, promoting T-cell exhaustion, as illustrated in Figure 7 [98]. There are currently two FDA-approved drugs indicated for the treatment of solid tumors with NTRK gene fusion, entrectinib, and larotrectinib, which act through selective inhibition of the tyrosine kinase domain of the NTRK [4]. Patients experiencing resistance to ICI therapy may see therapeutic benefits with concurrent administration of one of these small molecule inhibitors due to its attenuation of that JAK-STAT signaling pathway and

prevention of T-cell exhaustion, however, the repurposing of these drugs for treating ICI resistance remains an open avenue of investigation.

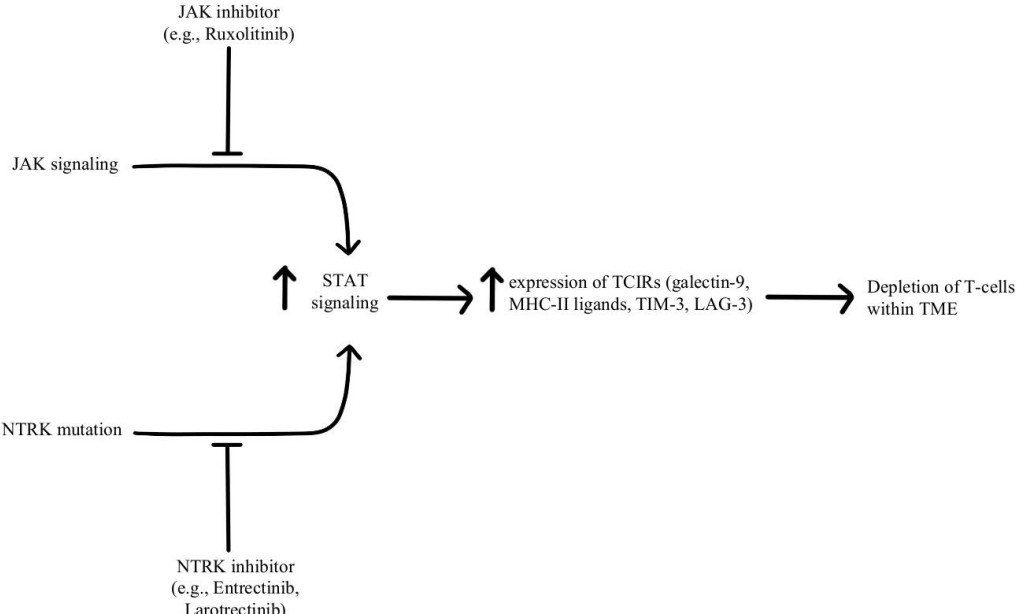

**Figure 7.** Diagram of STAT pathway modulation [4,96–98]. Illustrated by Kaitlyn Mi of The Dartmouth Institute.

### 4.3. Adoptive Cellular Therapy

Chimeric antigen receptor T-cell (CAR-T) therapy has been shown to achieve durable remissions in cancers and is currently FDA-approved for the treatment of certain hematological malignancies, and its combination with checkpoint inhibitors targeting PD-1 has provided successful outcomes in several cancers such as B-cell non-Hodgkin lymphoma, hepatocellular carcinoma, and metastatic melanoma [99–103].

In a safety and efficacy clinical trial, the use of anti-CD19 CAR-T cells in combination with nivolumab in 11 patients with relapsed/refractory non-Hodgkin lymphoma has shown that this combined treatment regimen is safe and mediates potent anti-lymphoma activity [104]. Another study utilized HER-2+ transgenic mice models to study the effects of CAR-T cells with and without PD-1 inhibition, and it was observed that the combination therapy improved the therapeutic efficacy of CAR-T-cells against solid cancers [105]. The synergy seen in the combination therapy is likely multifactorial, being a product of increased T-cell effector activity and other effects, such as enhanced T-cell survival [106].

Investigations into the gene editing of CAR-T cells have revealed that PD1-knockout CAR-T cells are a clinically efficacious alternative to combination therapy. Specifically, it was observed in murine models that editing the PD1 gene on tumor-specific T-cells to reduce the expression of PD-1 resulted in a significant delay in tumor growth following administration to PD-L1 overexpressing tumors [107,108]. Further data supports these findings, in that PD-1 deficient CAR-T cells showed significantly better antitumor efficacy than wild-type CAR-T cells [109–111].

In pursuit of more effective treatments, the engineering of CAR-T cells has been explored, such as the modification of CAR-T cells to secrete PD-1 blockers–a modification that was found to control tumor growth in lung cancer models significantly better than standard CAR-T cell therapy [112]. In a similar study looking at humanized mice models of renal cell carcinoma, carbonic anhydrase-targeting CAR-T-cells engineered to secrete PD-L1 antibodies were shown to combat T-cell exhaustion and increase recruitment of NK cells to the tumor tissue, while in another study CAR-T cells were engineered to express an anti-PD-1 single chain variable fragment, which was shown to attenuate the PD-1-dependent

inhibition of CAR-T cells as well as tumor-specific non-CAR-T cells, resulting in significant control of tumor growth when compared to wild type CAR-T-cells [113,114].

In addition, there are several ongoing clinical trials using CAR-T-cell therapy in combination with checkpoint blockers in the treatment of B-cell lymphoma, multiple myeloma, and Hodgkin lymphoma (NCT04134325, NCT04162119, NCT04163302, and NCT04213469). CAR-T-cell therapy and PD-1 inhibitors have both individually revolutionized the field of oncology, and their combination has been shown to be both synergistic and efficacious in overcoming resistance to checkpoint inhibition.

### 4.4. Tumor Neoantigen Vaccines

Heterogeneous response to PD-1/PD-L1 immunotherapy has prompted the investigation into strategies to overcome primary and acquired resistance. One such polypharmaceutical approach is the use of tumor neoantigen vaccines in combination with ICIs to potentiate the tumor-specific immune responses [115,116].

Tumor antigens, or neoantigens, are proteins displayed by tumor cells due to mutations in host DNA. Neoantigens stimulate CD4/8-MHC cell-mediated immune reactions, but clonal evolution promotes evasion of the normal response [117]. Studies support that tumor neoantigen vaccination is effective in individuals with an absence of pre-existing antitumor immunity. However, the TME expresses several immunosuppressive factors through prolonged exposure to cancer antigens, and/or poor infiltration of T-cells [118]. Anti-PD-1/PD-L1 immunotherapy has shown promising results in many solid tumors, with pancreatic cancer and glioblastomas being a few exceptions, possibly due to the lack of permeability of T-cells in tumor tissue [119]. A study outlined the following 3 distinct phenotypes describing the permeability of tumor cells based on epigenetic RNA N6-methyladenosine (m6A) modification in 1938 gastric cancer samples: immune-excluded, immune-inflamed, and immune-desert phenotypes. Based on their study, a low m6A score (indicating low levels of methylation and greater T-cell tissue infiltration) predicted an immune-inflamed phenotype, a better response to PD-1/PD-L1 immunotherapy, and a 5-year survival rate of 69.4% [120]. Cancer vaccines targeting neoantigens have been proposed as a tool to circumvent immunotherapy-induced resistance, increase T-cell permeability, and decrease T-cell exhaustion, potentially increasing the effectiveness of ICIs [118].

A Phase 1b feasibility study used the DC neoantigen vaccine in combination with anti-PD-1 therapy in pancreatic ductal adenocarcinomas to reprogram the TME and showed positive outcomes, a step forward in using these advanced therapeutic platforms, especially for treatment-refractory cancers [118]. Another vaccine approach currently in studies is using PD-1/PD-L1 checkpoint inhibitors with the GVAX vaccine–an anti-cancer vaccine consisting of irradiated tumor cells engineered to secrete granulocyte-macrophage colony-stimulating factor (GM-CSF) [121]. GM-CSF is a hematopoietic cytokine responsible for inducing the production of white blood cells, including granulocytes, macrophages, and megakaryocytes. GM-CSF also functions as a major pro-inflammatory cytokine, coordinating communication between lymphocytes and myeloid cells [122]. Another study combined PD-1/CTLA-4 checkpoint blockade with the GVAX vaccine in preclinical models of colon cancer and ovarian cancer with the eradication of tumor cells [123]. In addition to the GVAX vaccine, GM-CSF is used in the similar yet distinct treatment modality of oncolytic virotherapy, appearing as one of the principal genetic modifications in the only current FDA-approved oncolytic virus, Talimogene laherparepvec (T-vec), wherein GM-CSF likewise serves a proinflammatory role, potentiating an anti-tumor immune response. T-vec has shown clinical efficacy in treating metastatic melanoma, for which it received its original FDA approval, while recent investigations into its combination with PD-1 immunotherapy have shown promising clinical benefits [4]. This success has further galvanized interest in the combination of oncolytic viruses with PD1 inhibitors, with the field seeing an explosion in the diversity of viruses, spanning both DNA and RNA viruses, used in combination with PD1 checkpoint therapy [124].

Several high-powered clinical trials are currently investigating the safety and efficacy of tumor neoantigen vaccines in combination with PD-1/PD-L1 immunotherapy, radiotherapy, and conventional chemotherapy in the treatment of tumors of both solid and hematogenous origin. A phase 2 trial concluding in 2019 assessed the safety and efficacy of the GVAX vaccine and cyclophosphamide combined with or without nivolumab. In total, 93 enrolled subjects with metastatic pancreatic adenocarcinoma were given the GVAX/cyclophosphamide with or without anti-PD-1 immunotherapy. After the conclusion of the trial, the overall survival was 5.88 months for the (+) nivolumab group (95% CI = 4.73–8.64), while it was 6.11 months for the (−) nivolumab group (95% CI = 3.52–7.00) (NCT02243371). More recently, another study used modified DC vaccines combined with nivolumab to treat six patients with malignant gliomas. Subjects receiving neoadjuvant therapy experienced a median progression-free survival of 4.3 months with a 95% confidence interval of 2.1–5.3 months, while those in the adjuvant therapy group had a median progression-free survival of 6.3 months with a 95% confidence interval of 4.7–10.7 months (NCT02529072). Radiotherapy with ICIs and/or vaccination may prove useful in maximizing the tumor microenvironment and treatment response. A study published by Zhang et al. found that combination E7 vaccination-radiotherapy proved efficacious in altering the ratio of CD8+ T cells/Tregs, increasing the number of CD45+CD8+ cells and effector CD8+ IFN-γ+ T cells, and decreasing overall tumor weight as compared to conventional monotherapy or other studied combinations [125]. Outside of these two trials, there are over 100 active and recruiting studies, indicating the available literature on this approach to overcoming PD-1/PD-L1 resistance will likely increase dramatically going forward.

*4.5. Gut Microbiota*

Another active field of study in overcoming ICI resistance is that of the gut microbiome and its manipulation. The gut microbiome is known to directly regulate the classically studied and well-understood gastrointestinal processes such as the absorption, metabolism, and excretion of nutrients and pharmaceutical agents, but it has further been found to regulate more complex processes such as the maintenance of the intestinal mucosal barrier, the synthesis of short-chain fatty acids, the modification of the regulation of key hormones and neurotransmitters such as CCK, GLP-1, and 5-HT, and host immunity [126–129]. When the homeostasis of the microbiome is disturbed due to antibiotics or some other destructive exposure (through environment, diet, lifestyle, smoking, alcohol, etc.), the sensitive balance between commensals, pathobionts, and pathogens can become unsettled and cause considerable pathology in the patient. If not self-resolved or remedied by medication, such disturbances in the microbiome can cause significant metabolic derangements in addition to initiating chronic inflammatory processes and inflicting permanent damage to the underlying mucosa [128,130]. The literature describes the relationship between the microbiome and drugs as "bidirectional," due to their ability to both positively and negatively affect the functions and efficacy of one another [131]. Manipulation of the gut microbiome may therefore possess beneficial clinical applications and should continue to be assessed as a possible method of overcoming PD-1/PD-L1 blockade resistance.

Current evidence from murine models supports this hypothesis, showing that the gut microbiome plays a significant role in the clinical response to ICIs targeting the PD-1/PD-L1 axis. A study observed decreased efficacy of ICIs in germ-free mice and mice pretreated with antibiotics. After the manual inoculation of bacterial species in these mice, the use of ICIs led to significantly higher T-cell proliferation. Lastly, feces from ICI-responsive patients were transferred into mice, which showed increased efficacy of ICI immunotherapy, providing evidence to suggest fecal microbiota transplant (FMT) as a viable combination therapy to potentiate ICI efficacy [132]. Another study discovered that the use of Bifidobacterium species enhanced the efficacy of PD-L1 therapy [133]. Aside from murine models, numerous studies in the literature have retrospectively and prospectively assessed the content of microflora among human patients receiving PD-1/PD-L1 therapy

for metastatic melanoma, hepatocellular carcinoma, gastrointestinal cancer, NSCLC, and several other diseases [131,134–140]. Heterogeneous response rates in ICI-induced resistance in cancers due to the diverse gut microbiome are an active area of investigation. Long-term data may serve to guide clinical decision-making regarding FMT and/or bacterial supplements concurrently with PD-1/PD-L1 therapy. Currently, 12 plus relevant trials are investigating the effect of microbiome alteration, either through FMT or probiotic supplementation, on the efficacy of anti-PD-1 therapy. A phase 1b study was designed to assess the safety and efficacy surrounding SER-401 FMT and nivolumab therapy in 14 patients between 2019 and 2022 who received pre-treatment with vancomycin or placebo followed by SER-401 or placebo and nivolumab; results are in progress (NCT03817125). Another ongoing clinical study aims to understand whether FMT can convert a patient's ICI-refractory metastatic melanoma to ICI-responsive. These study patients will receive FMT from either ICI-responding or non-responding patients alongside their standard immunotherapy to investigate (NCT05251389). Meanwhile, another trial is assessing the feasibility of an FMT capsule with ICI therapy in treatment-refractory gastrointestinal cancers (NCT04130763). Other ongoing studies are assessing the use of FMT, prebiotics, and/or probiotics alongside ICI therapy for several cancer types, including NSCLC, hepatocellular carcinoma, and gastric cancer (NCT05008861, NCT03772899, NCT04101747, NCT04130763, NCT04924374, NCT04988841, NCT05001360, NCT05032014, NCT05094167, NCT05251389, and NCT05303493). More studies are required to modulate the gut microbiome in ICI-refractory cancers to maximize clinical efficacy.

## 5. Discussion

Monoclonal antibodies targeting PD-1/PD-L1 have demonstrated clinical benefits in several cancers, including melanoma, NSCLC, renal cell cancer, bladder cancer, head and neck cancers, and MSI-high colorectal cancer to name a few [141,142]. Long-term follow-up of patients in clinical trials utilizing ICIs has identified the following three broad groups: responders, non-responders, and responders that fail to respond over time [143]. Challenges remain in understanding the differences between responders and non-responders given intra- and inter-patient heterogeneity. Additionally, incomplete knowledge of clinical, molecular, and immunological mechanisms driving innate and acquired resistance poses a major problem. Our understanding about the PD-1/PD-L1 axis on immune cells and cancer cells is ever-evolving. Tumor-specific CD8+T-cells differentiate from effector T-cells, undergo clonal expansion, infiltrate TME, and aid in tumor killing by displaying tumor-associated antigens [115]. Lack of anti-tumor responses with ICI therapy can result from the suboptimal generation of anti-tumor T cells, the inadequate function of tumor-specific T-cells, and impaired memory T-cells [144,145]. Another set of mechanisms creating resistance to ICIs includes a lack of suitable neoantigen, dysfunctional neoantigen processing, and impaired presentation of neoantigen to form tumor-specific T-cells [115]. Depletion, exhaustion, and dysfunction of CD8+T-cells can arise through diverse immunosuppressive components of TME [146].

Immune evasion is a multifactorial process that includes genetic/epigenetic aberrations that lead to insufficient neoantigen presentation, malfunctional presentation, and further processing that disrupts the cytotoxic potential of CD8+T-cells [146]. The presence of non-cancerous cells or diverse host microbiota can also promote ICI resistance in tumors [147]. Anti-PD-1/PD-L1 blockers reactivate these tumor-specific T-cells. However, altered neoantigen production and processing can impair anti-tumor immune responses [148]. The presence of high levels of non-synonymous mutations in cancers leads to the highest response rates to ICIs. However, mutations in genes encoding antigen processing components such as B2M result in defective MHC-I presentation and hence suboptimal response to ICIs [149,150]. Ever changing mutational landscape and clonal expansion of neoantigens have been responsible for acquiring resistance to ICIs. A key strategy to improve immunogenic cell death is to stimulate innate immunity and DC functionality such as type 1 interferons, TLR ligands, oncolytic viruses, etc. that can promote

neoantigen production in TME [146]. Specific oncogenic signaling such as loss of PTEN diminishes T-cell infiltration; alterations in wnt signaling lead to DC trafficking in TME; an oncogenic KRas mutation is associated with higher levels of T-cell exhaustion; the presence of immunosuppressive gene signature, they all lead to multigenic reversible pathways of resistance [83,151,152]. It is plausible to overcome ICI resistance by altering these intrinsic and extrinsic signals causing PD-1/PD-L1 resistance.

Increased TMB is associated with increased PD-L1 expression as seen in advanced melanoma, resulting in improved progression-free survival and overall survival when treated with combined or mono-immunotherapy versus patients with negative PD-L1 [153]. However, there are cancers with equally increased PD-L1 expression, yet no effective immune response from ICI, making PD-L1 not a very reliable biomarker for the desired responses from checkpoint blockade [154]. Moreover, inhospitable TME can threaten expanding the repertoire of anti-tumor T-cells because of high levels of alternate checkpoints of co-inhibitory receptors on T-cells (CTLA-4, TIM-3, and LAG-3), high levels of immune suppressive cytokines, and recruitment of MDSCs as well as Treg cells [115]. These cellular processes led to inadequate anti-tumor T-cell effector functions. Moreover, mutations in immune effector signaling pathways such as JAK1/JAK2 are associated with resistance to PD-1 blockade [150]. These mutations resulted in a loss of interferon responsiveness. Similarly, mutations in B2M, deletion of IFN-$\gamma$ receptors, and mutations in components of the JAK/STAT pathway contribute to resistance to PD-/PD-L1 blockade [155]. Importantly, these signaling pathways can be inhibited and could overcome acquired resistance to ICI blockers.

The heterogeneity of PD-1+CD8+T-cells in TME responds differently to anti-PD-1 treatment [156]. Exhausted PD-1+CD8+T-cells display a unique genetic landscape compared to effector T cells [157]. These epigenetic regulators in exhausted and non-exhausted T cells drive variable responses to checkpoint blockade. One such TIL subpopulation, TCF1+PD-1+CD8+ T-cells with stem-like characteristics, has been identified as a primary mediator of the therapeutic response to PD1 checkpoint inhibition. This population proliferates in response to immunotherapy, forming terminally differentiated TCF1-PD1+CD8+ T-cells in addition to increasing the progenitor TCF1+PD-1+CD8+ population. Studies, such as that of Siddiqui, et al., have shown that ablation of this population restricts responses to PD1 checkpoint immunotherapy, lending further credence to the thought of modulating T-cell populations to potentiate checkpoint inhibitor efficacy [158–160]. In addition to checkpoint inhibitor monotherapy, co-inhibitory receptors can be blocked with combination checkpoint blockade using LAG-3+PD-1 and TIM-3+PD-1, which have shown better response rates in pre-clinical models as well as currently being studied in clinical trials [161]. PD-1/PD-L1 independent mechanisms of immune evasion such as Tregs, MDSCs, Th2 CD4+ T cells, and M2-polarized tumor-associated macrophages collectively influence the resistance to anti-PD-1/PD-L1 treatments. Removal of anti-PD-1 antibodies to inhibit its interaction with PD-1+CD8+T-cells via an Fc$\gamma$R-dependent manner has been reported [162]. Pre-clinical studies have been successful in eliminating this alternate population of immune cells that are driving the innate resistance to ICIs. In most tumors, the formation of memory T-cells as a response to either tumor formation or immune checkpoint therapies is impaired. These processes lead to a lack of durable responses. The transcriptional machinery in TME defines the fate of these T-cells in naïve, memory, and exhausted T-cell states [162]. Ongoing clinical studies are augmenting existing new T-cell responses or priming new populations of T-cells using adoptive cellular therapy like CAR-T cells.

Even if the T-cell activation process were not defective, there could be a disruption in T-cell trafficking and tumor infiltration because of the downregulation of chemokines such as CXCR3 required for T-cell recruitment [163]. Epigenetic regulation in silencing CXCL9 and CXCL10 in cancers leads to chemokine repression and tumor progression. T-cells traffic in lymph nodes by endothelin B receptors, and genetic changes in ETBR result in a potential immune escape mechanism [164]. Overexpression of VEGF downregulates T-cell adhesion, TILs, and shorter overall survival as a response to ICI. TME plays a crucial role by inducing

the expression of PD-L1 and Tregs through several oncogenic signals, thereby decreasing cytotoxic T-cell functionality [165].

In summary, resistance to immune checkpoint inhibitors (PD-1/PD-L1) is driven by complex genetic/epigenetic interactions between immune cells and tumor cells. These innate and acquired resistance pathways overlap. Ongoing clinical trials are combining immunotherapeutic agents with targeted agents, cytotoxic chemotherapy, and/or radiation, all in an effort to provide a durable response to these revolutionary therapies. Alternative strategies, such as cancer neoantigen vaccines, CAR-T cellular therapies, genotype-targeted therapies, and modulation of the gut microbiome, are required to overcome resistance in non-responders. It is prudent to identify unique biomarkers, both predictive and prognostic, in responders vs. non-responders to ICIs. Checkpoint inhibitor immunotherapy has highlighted the possibility of curing cancers, yet a significant unmet clinical need is to be able to understand the landscape of resistance to ICI early on to customize therapies based on every patient's unique tumor/immune genetic signature.

**Author Contributions:** Conceptualization, J.M. (John Moise) and S.A.; investigation, J.M. (John Moise), J.M. (Jeevan Murthy), D.D., S.Y., F.K., E.H. and S.A.; writing—original draft preparation, J.M. (John Moise), J.M. (Jeevan Murthy), S.Y. and S.A.; writing—review and editing, J.M. (John Moise); supervision, S.A.; project administration, J.M. (John Moise); funding acquisition, S.A. All authors have read and agreed to the published version of the manuscript.

**Funding:** This research received no external funding.

**Acknowledgments:** The authors would like to extend their deepest gratitude to Kaitlyn Mi of The Dartmouth Institute for her illustrations of the biochemical pathways in Sections 4.1 and 4.2.

**Conflicts of Interest:** The authors declare no conflict of interest.

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
