# Peer review of "Mechanisms of Resistance and Strategies to Combat Resistance in PD-(L)1 Blockade"

_2673-5601, doi:10.3390/immuno2040041_

Round 1

Reviewer 1 Report

I have. no concerns about this and recommend it be published.  I do however have a minor comment that may warrant an additional line or two:

Whilst this review focuses on the major pathways and biology of the PD1/PDL-1 interaction one point not highlighted is that PDL2 can also bind PD-1 and may be present in many tumours.  This may be an additional and under-researched mechanism into PD1/PDL-1 resistance and warrants a mention, especially in cases where targeting PDL-1 rather than PD-1 may fail.  I feel that highlighting a potential role for PDL-2 in PDL-1 blockade escape is an important factor to consider and could lead to clinical differences between PD-1 and PDL-1 antibody therapy. 

I want to take the opportunity to thank the authors for their review, I found it informative and well-written.

Author Response

I feel that highlighting a potential role for PDL-2 in PDL-1 blockade escape is an important factor to
consider and could lead to clinical differences between PD-1 and PDL-1 antibody therapy.

a. Section discussing the importance of PD-L2 was added to lines 115-122

Reviewer 2 Report

Revision of the review : “ Mechanisms of resistance and strategies to combat resistance in PD-(L)1 blockade”.

The review “ Mechanisms of resistance and strategies to combat resistance in PD-(L)1 blockade” is an interesting review that discusses about different aspect of the resistance in PD-(L)1 blockade treatment against cancer. The review begins by describing PD-(L)1 expression and function followed by the therapeutic application of immune checkpoint inhibitors against PD-(L)1 for tumor treatment. The following chapter describes the different mechanisms that can lead to PD-(L)1 resistance and proposes approaches that might allow to overcome such mechanisms. The last chapter, the authors discuss about the different aspect of PD-(L)1 resistance and how those could be solved.

The review is easy reading and I believed that it is of interest for the scientific community. Thus I would recommend its publication after addressing the minor but important revisions hereafter indicated.

Minor considerations.

IMPORTANT Consideration:

Although the review is well written, it becomes hard to follow as the authors did not include figures to help/summary the different mechanism of  PD-(L)1 resistance and strategies of overcome it. I strongly recommend the introduction of some figures.

-line 25: Please check that in the below text you use the abbreviation ICIs and not ICIs and “immune checkpoint inhibitor” word.

-line 61: Please indicate the name for the abbreviation SHP.

-line116: Can you please provide some statistical values for the resistance observed.

-line 130: Please indicate how those immunosuppressive cytokines could be targeted? Using monoclonal antibodies in combination with ICI?

-line 227: Please indicate the name before as this abbreviation was used previously in the text.

-line 264: Is there any research done in order to propose a "solution" to prevent NTRK action?

-line 270: Please provide some examples of cancers.

-line 312: Please described better what is neoplastic cells, most of the reader might not know.

Author Response

1. line 25: Please check that in the below text you use the abbreviation ICIs and not ICIs and “immune
checkpoint inhibitor” word.
a. The two subsequent uses of “immune checkpoint inhibitors” were replaced with “ICIs” in
lines 92 and 187.
2. line 61: Please indicate the name for the abbreviation SHP.
a. The full name of SHP was added to lines 61-62.
3. line 116: Can you please provide some statistical values for the resistance observed.
a. An estimation for the number of patients with primary and secondary therapy resistance
was added to lines 127-128.
4. line 130: Please indicate how those immunosuppressive cytokines could be targeted? Using
monoclonal antibodies in combination with ICI?
a. Yes, concurrent administration of anti-cytokine antibodies and ICIs is the prevailing
methodology in the literature. Clarifying language was added to line 144.
5. line 227: Please indicate the name before as this abbreviation was used previously in the text.
a. Name was removed from line 252 in favor of abbreviation as the full name of SHP was
added to lines 61-62 (Edit #2).
6. line 264: Is there any research done in order to propose a "solution" to prevent NTRK action?
a. There are two FDA-approved drugs for inhibiting NTRK, although there has not been any
investigation into their use in preventing ICI resistance. An addendum discussing this
was added in lines 295-301.
7. line 270: Please provide some examples of cancers.
a. Example cancers were added to lines 306-307.
8. line 312: Please describe better what neoplastic cells are, most of the reader might not know.
a. “Neoplastic” was replaced by “tumor” for clarity on line 353

Reviewer 3 Report

This minireview about the mechanisms of resistance and associated strategies to overcome them is well written and documented.

However, several (minor) points need to be improved:

-in paragraph 4.1: the introduction concerning immunosuppressive cytokines suggests that only Tregs produce them, but this is not true, and even if one understands it later in the reading, this point should be clarified from the beginning so that the origin of cytokines in the tumor microenvironment is explicit.

-At the end of the first paragraph of 4.2 section, the CTLA4 molecule and anti-CTLA4 are discussed, but James Allison is the co-recipient of the Nobel Prize for the discovery of the Immune-Checkpoint, as is Tasuku HONJO, so it seems logical to me to highlight this point again in this paragraph.

- in the second paragraph of 4.2section : the authors talk about TIM-3, but they never mention its main ligand which is Galectin-9, whose blocking by a specific antibody is currently in clinical trials.

-In the 4th paragraph of 4.2: the link between B2-microglobulin and BempegAldesleukin mutations is not obvious, it is necessary to make 2 different parts.

- Concerning part 4.4 section on Neo-Antigens: targeted antigens should be discussed more regularly and specifically. It would also be necessary to discuss the vaccination techniques by ONCOLYTIC viruses which are themselves based on the use of GM-CSF and approved by the FDA. In addition, a short paragraph on combination trials with conventional therapies (chemotherapy, radiotherapy) of varying regimens would be more comprehensive. 

- In this same paragraph, it is noted on line 358 that vaccination with neo-antigens is a new approach, but it is more a question of a renewed interest in this approach, which is indeed one of the ways to overcome resistance to ICI immunotherapy.

- In the 4th paragraph of the discussion (5.), the authors discuss TCD8+, but a specific phenotype of TCD8+: PD1+ TCF1+ stem-like CD8+ T cells, have been identified as correlating with anti-PD1 response. This should be addressed.

Finally, it seems important to me, that a mini-review includes a summary figure of the major data it puts forward. Therefore, I strongly recommend the creation of a simplified figure with a caption in order to provide a clear picture of the key points discussed.

Author Response

1. in paragraph 4.1: the introduction concerning immunosuppressive cytokines suggests that only Tregs produce them, but this is not true, and even if one understands it later in the reading, this point should be clarified from the beginning so that the origin of cytokines in the tumor microenvironment is explicit.
a. Language on lines 140-141 was modified to eliminate any implication that Treg cells are the only source of immunosuppressive cytokines.
2. At the end of the first paragraph of 4.2 section, the CTLA4 molecule and anti-CTLA4 are discussed, but James Allison is the co-recipient of the Nobel Prize for the discovery of the Immune-Checkpoint, as is Tasuku HONJO, so it seems logical to me to highlight this point again in this paragraph.
a. Credit to James Allison was added in lines 219-220.
3. in the second paragraph of 4.2section : the authors talk about TIM-3, but they never mention its main ligand which is Galectin-9, whose blocking by a specific antibody is currently in clinical trials.
a. A sentence discussing the LYT200 clinical trial was added to lines 243-246.
4. In the 4th paragraph of 4.2: the link between B2-microglobulin and BempegAldesleukin mutations is not obvious, it is necessary to make 2 different parts.
a. Clarifying language was added to lines 265-270.
5. Concerning part 4.4 section on Neo-Antigens: targeted antigens should be discussed more regularly and specifically. It would also be necessary to discuss the vaccination techniques by ONCOLYTIC viruses which are themselves based on the use of GM-CSF and approved by the FDA. In addition, a short paragraph on combination trials with conventional therapies (chemotherapy, radiotherapy) of varying regimens would be more comprehensive.
a. Section discussing the use of oncolytic viruses in combination with PD1 checkpoint therapy was added to lines 385-395.
b. Section discussing combination trials with conventional therapies (chemotherapy, radiotherapy) of varying regimens was added to lines 411-416.
6. In this same paragraph, it is noted on line 358 that vaccination with neo-antigens is a new approach, but it is more a question of a renewed interest in this approach, which is indeed one of the ways to overcome resistance to ICI immunotherapy.
a. Language was changed to reflect this in lines 349 and 417.
7. In the 4th paragraph of the discussion (5.), the authors discuss TCD8+, but a specific phenotype of TCD8+: PD1+ TCF1+ stem-like CD8+ T cells, have been identified as correlating with anti-PD1 response. This should be addressed.
a. Section on TCF1+PD1+ stem-like CD8+ T-cells was added to lines 533-541.
8. Finally, it seems important to me that a mini-review includes a summary figure of the major data it puts forward. Therefore, I strongly recommend the creation of a simplified figure with a caption in order to provide a clear picture of the key points discussed.
a. A summary figure has been created for inclusion with the mini-review.